# Considerations for Distribution Shift Robustness of Diagnostic Models in Healthcare

## Abstract

We consider robustness to distribution shifts in the context of diagnostic models in healthcare, where the prediction target $Y$, e.g., the presence of a disease, is causally upstream of the observations $X$, e.g., a biomarker. Distribution shifts may occur, for instance, when the training data is collected in a domain with patients having particular demographic characteristics while the model is deployed on patients from a different demographic group. In the domain of applied ML for health, it is common to predict $Y$ from $X$ without considering further information about the patient. However, beyond the direct influence of the disease $Y$ on biomarker $X$, a predictive model may learn to exploit confounding dependencies (or shortcuts) between $X$ and $Y$ that are unstable under certain distribution shifts. In this work, we highlight a data generating mechanism common to healthcare settings and discuss how recent theoretical results from the causality literature can be applied to build robust predictive models. We theoretically show why ignoring covariates as well as common invariant learning approaches will in general not yield robust predictors in the studied setting, while including certain covariates into the prediction model will. In an extensive simulation study, we showcase the robustness (or lack thereof) of different predictors under various data generating processes. Lastly, we analyze the performance of the different approaches using the PTB-XL dataset, a public dataset of annotated ECG recordings.

## 1 Introduction

As predictive machine learning models are being increasingly relied upon in high-stakes domains, understanding their robustness under distribution shifts is crucial. In this work, we discuss this aspect in the context of the diagnostic setting in healthcare, where the goal is to diagnose the presence of disease $Y$ given a reading of a relevant biomarker $X$. Importantly, this setting implies that the target of the prediction $Y$ is causally upstream from the covariates $X$, which is why in the machine learning literature, this setting is commonly referred to as *anti-causal* (Schölkopf et al., 2012). Additionally, we may have access to a patient's metadata $V$, e.g., age or body mass index (BMI), which often also affect the biomarker as well as the disease prevalence. In this case, $V$ gives rise to confounding dependencies between $X$ and $Y$ which may change under certain distribution shifts. If a predictive model that only takes the biomarker $X$ as input learns to exploit such dependencies, often referred to as "shortcuts", performance may deteriorate under distribution shifts. Distribution shifts may occur, for instance, when the training data is collected in a domain with patients having particular demographic characteristics while the model is deployed on patients from a different demographic group, a common issue that needs to be considered in most applications of machine learning for healthcare.

### 1.1 Related work

Recently, there has been growing interest in studying the robustness of machine learning models under distribution shifts (e.g. Quionero-Candela et al. (2009); Storkey (2009); Koh et al. (2021); Geirhos et al. (2020)). One way to circumvent the adverse effects of distribution shift is in a data-driven way, by learning representations invariant to the shift by training the model on data coming from multiple "environments"

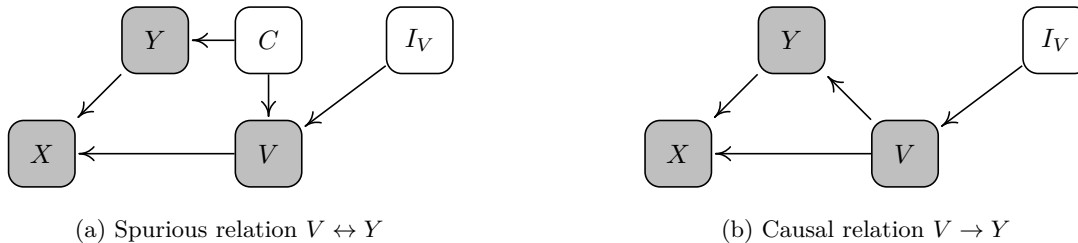

(a) Spurious relation $V \leftrightarrow Y$          (b) Causal relation $V \rightarrow Y$

Figure 1: Data generating processes considered in this work. $I_V$ is an intervention variable, which describes the assumed distribution shift. **(a)** The "spurious relation process" features a spurious relation between $Y$ and $V$, through a confounding variable $C$, and has been considered in the ML literature (Heinze-Deml & Meinshausen, 2021; Veitch et al., 2021; Makar et al., 2022; Puli et al., 2022). This setting requires that the marginal $P(Y)$ remains invariant across distribution shifts. **(b)** In the "causal relation process", the shortcut variable $V$ is a direct cause of the outcome $Y$, shifting the marginal $P(Y)$ when $I_V$ shifts the marginal $P(V)$.

or "domains" (Ganin et al., 2016; Peters et al., 2016; Heinze-Deml et al., 2018; Rothenhäusler et al., 2021; Arjovsky et al., 2019; Magliacane et al., 2018; Krueger et al., 2021) where the relevant distribution shifts are already present in some form. The caveat of this approach is that it requires training data from a range of distribution shifts, which might not always be available.

In settings without annotated training data from various environments, but access to domain knowledge, another line of work focuses on modelling the full data generating mechanism along with the expected distribution shifts to derive modelling strategies that ensure robustness under such shifts (Pearl & Bareinboim, 2011; Subbaswamy et al., 2019; Subbaswamy & Saria, 2018; Subbaswamy et al., 2022; Heinze-Deml & Meinshausen, 2021; Veitch et al., 2021; Makar et al., 2022; Puli et al., 2022). Recently, there has been a special focus on a particular kind of spurious relation between covariate $V$ and $Y$ as depicted in Fig. 1a[1]. However, in many applications of interest—such as the diagnostic setting in health—there can exist not only spurious relations between $V$ and $Y$ (Fig. 1a), but also ones where $V$ has causal influence on $Y$ (Fig. 1b)[2].

One illustrative example for a causal relationship between $V$ and $Y$ is that the body mass index (BMI) is causally related to a host of conditions, e.g., left ventricular hypertrophy (LVH) (Lorell & Carabello, 2000). BMI is not "spurious" in the sense that it is merely associated with LVH, but can directly cause changes in left ventricular mass, which in turn can lead to LVH (Himeno et al., 1996). However, a shift in the prevalence of elevated BMI can shift the association between a signal—e.g., an electrocardiogram (ECG)—that is influenced by both BMI and LVH.

In the literature on applications of ML to healthcare diagnostics, many works do not take advantage of additional covariates $V$ when designing their predictors, and use solely the key biomarker $X$ without an explicit explanation for this design choice. For example, there is a long line of work using machine learning to predict cardiovascular diseases from the ECG signal without considering usage of the available auxiliary covariates such as age, BMI, or sex (Strodthoff et al., 2021; Mehari & Strodthoff, 2022; Śmigiel et al., 2021; Kiyasseh et al., 2021; Nonaka & Seita, 2021; Attia et al., 2019b;a; Galloway et al., 2019; Hannun et al., 2019), while there exists evidence from the medical literature that the underlying process follows Fig. 1b (Rodgers et al., 2019; Lorell & Carabello, 2000; Himeno et al., 1996). In other diagnostic problems, e.g., diagnosing diseases from chest X-rays, we find the same pattern in the literature where the available auxiliary covariates are ignored (Rajpurkar et al., 2017; Suriyakumar et al., 2023; Adam et al., 2022; Jiao et al., 2021). This array of applied work in the health domain shows that it is not standard practice to include additional covariates such as demographics into prediction models, but instead many works choose to *not include* certain or any of them. In this work, we highlight that this can lead to unrobust predictors whose performance can be negatively affected by distribution shifts around these additional covariates, as summarized below.

---

[1]A similar problem setting is considered by the fairness literature (Kilbertus et al., 2017; Kusner et al., 2017).
[2]Storkey (2009) considers similar settings, where $X$ causes $Y$, under the names of *source component shift* and *mixture component shift*.

### 1.2 Contributions

We analyze the possible consequences of omitting certain covariates on the performance of predictive models under distribution shift in each of the settings in Fig. 1. To that end, we leverage recent theoretical results from the causality literature (Pfister et al., 2021; Subbaswamy et al., 2022) and provide an extensive simulation study on a synthetic data generating process. We find that in the causal relation setting, (Fig. 1b), learning predictors based on representations invariant to auxiliary covariates $V$, or not using those covariates at all, can result in predictors that lack robustness with respect to shifts in the distribution of $V$. The solution that achieves the optimal robustness is to include the auxiliary covariates $V$ as inputs to the predictor. On the other hand, our simulation study also highlights that including $V$ in the spurious relation setting (Fig. 1a) can make predictors less robust than simply ignoring $V$—in this setting invariant learning techniques should still be preferred. This highlights the importance of determining which of the settings the problem under consideration belongs to. Lastly, we consider PTB-XL (Wagner et al., 2020), a publicly available dataset of annotated ECG recordings, and we outline a procedure one can use to diagnose which of the settings applies, in cases when data from the target domain is available.

With this work, we aim to didactically highlight the difference between the two settings, draw the attention of the community to this phenomenon, and reconsider the common design choices in this area—we advise that ML practitioners in the healthcare domain carefully model the data generating process to assess whether including auxiliary covariates is appropriate in their setting.

## 2 Problem Setting

Consider predicting the outcome $Y$ (e.g., the presence of a disease) from a biomarker $X$ (e.g., an ECG recording) in the presence of an auxiliary covariate $V$ (e.g., age or BMI), in a setting where the data generating process between $Y$ and $X$ is anti-causal, i.e., where $Y$ has a causal effect on $X$. We also assume $V$ has a causal effect on $X$. The two considered settings in Fig. 1 differ in their relationship between $V$ and $Y$: in Fig. 1a there is a spurious relation between $V$ and $Y$, while in Fig. 1b $V$ has a causal effect on $Y$. We consider distribution shifts that occur under interventions on $V$ (e.g. shifts in the age or BMI distribution). These distribution shifts are indicated via the intervention variable $I_V$ (Pearl, 2009). In each setting, the goal is then to seek a predictor that performs well across the class of distributions that are generated by the interventions which we contrast below:

**Spurious relation** $V \leftrightarrow Y$   In the setting illustrated by the graph in Fig. 1a, the goal is to develop a predictor that is robust to shifts across the family of related probability distributions

$$\mathcal{P}_{spur} = \{P_s(X|Y,V)\,P_s(Y)\,P_t(V|Y)\}, \tag{1}$$

for a source distribution, denoted by $s$, and shifted target distributions, indexed by $t \in \mathcal{T}^{spur}$ as in Makar et al. (2022). All target distributions in this family of distributions thus factor as $P_t(X,Y,V) = P_s(X|Y,V)\,P_s(Y)\,P_t(V|Y)$, i.e., they vary from the source distribution only in $P(V|Y)$, while $P(X|Y,V)$ and $P(Y)$ remain unchanged. Notably, the assumption that $P(Y)$ remains the same across all potential shifted distributions can be unrealistic in applications like healthcare. For example, we would expect the prevalence of heart diseases ($Y$) to be higher in an older population ($V$).

**Causal relation** $V \rightarrow Y$   If, instead, the auxiliary covariate $V$ is a direct causal parent of the outcome $Y$, as depicted in Fig. 1b, we wish to be robust to distribution shifts inside the following family of distributions, indexed by $t \in \mathcal{T}^{cause}$:

$$\mathcal{P}_{cause} = \{P_s(X|Y,V)P_s(Y|V)P_t(V)\}. \tag{2}$$

where we allow for a changing marginal distribution of $P(V)$, while holding the conditional distributions $P(Y|V)$ and $P(X|Y,V)$ fixed.

Methods introduced in Pfister et al. (2021); Subbaswamy et al. (2022) allow to derive stability properties of predictors from the causal graph, relying on d-separation (Pearl, 2009). In the following, we first apply

the notion of *stable sets* from Pfister et al. (2021) before also providing a direct derivation for readers not familiar with d-separation.

Exploiting the notion of *stable sets*, we can derive which sets of predictors are associated with the same conditional distribution of $Y$ across different interventions on $V$ by observing which sets of covariates d-separate $I_V$ and $Y$ in the graph in Fig. 1b. All such stable sets must contain the covariate $V$ to d-separate $I_V$ and $Y$ (as the path $I_V \rightarrow V \rightarrow Y$ needs to be blocked (Pearl, 2009)). The predictive distribution derived from the source distribution that conditions on $X$ and $V$ is then invariant across the entire family, i.e., $P_s(Y|X,V) = P_t(Y|X,V)$, whereas the predictive distribution that only conditions on $X$ is, in general, not invariant, i.e., $P_s(Y|X) \neq P_t(Y|X)$. Similarly, since the empty set is not stable, we have that $P_s(Y) \neq P_t(Y)$ in general.

We summarize this insight in the following proposition and provide a direct derivation for readers not familiar with the graphical notions used above in Appendix A.

**Proposition 1** *For any element $P_t \in \mathcal{P}_{cause}$ as defined in Eq. (2), it holds that $P_t(Y|X,V) = P_s(Y|X,V)$. Furthermore, for such a $P_t$, in general $P_t(Y|X) \neq P_s(Y|X)$ as well as $P_t(Y) \neq P_s(Y)$ .*

Hence, in a scenario in which the data generating mechanism from Fig. 1b applies, $\{V, X\}$ should be used as input to the predictive model and a predictor using only $\{X\}$ would not be robust to the considered distribution shifts. Finally, the assumption of $P_t(Y) = P_s(Y)$ as required by Makar et al. (2022) does not hold for $\mathcal{P}_{cause}$ in general.

**Remark 1** *Proposition 1 generalizes to an arbitrary (finite) number of auxiliary covariates $V_1, \ldots, V_m$, as long as they each cause both $Y$ and $X$, in a way such that the resulting distribution shifts are of the form $P^m_{cause} = \{P_s(X|Y, V_1, \ldots, V_m) P_s(Y|V_1, \ldots, V_m) P_t(V_1, \ldots, V_m)\}$. The proof is analogous to the proof of Proposition 1. Note that this does not assume any particular relationship between the different $V_i$, i.e. their joint $P_t(V_1, \ldots, V_m)$ can factor in an arbitrary way for this to hold.*

## 3  Simulation study

To illustrate our theoretical claims about the consequences of distribution shift in a data generating process with causally influencing covariates $V$, we set up a synthetic model:

$$P(V = 1) = p, \qquad\qquad P(Y = 0|V = 0) = p_0,$$
$$P(X|Y = y, V = v) = \mathcal{N}(\mu_{y,v}, 1), \qquad P(Y = 1|V = 1) = p_1. \tag{3}$$

This model is constructed to allow to analyze the shifts of the family $\mathcal{P}_{cause}$, where $P(V)$ can be shifted by varying $p$, while $P(Y|V)$ and $P(X|Y,V)$ remain fixed. We obtain $P(Y|X)$ and $P(Y|X,V)$ in closed form, which allow us to form predictors with $\arg\max_y P(Y = y|\cdot)$—we will refer to these as "predictor $P(Y|\cdot)$". The performance metrics, area under the receiver operating curve (AUC) and accuracy, are estimated based on $2^{16}$ samples drawn from the target joint distribution $P_t(X, Y, V)$.

Throughout this section, we will compare the performance of two distinct kinds of predictors: target and source predictors, using distributions $P_t(Y|\cdot)$ and $P_s(Y|\cdot)$, respectively. The target predictors $P_t(Y|\cdot)$ are oracle predictors: they are obtained from the target joint distribution $P_t(X, Y, V)$ itself, and so they are not influenced by the distribution shift. Hence, the performance of $P_t(Y|\cdot)$ will be the upper bound on the distribution shift robustness of any predictor conditioned on the same information. Nevertheless, the performance metrics of predictors $P_t(Y|\cdot)$ will change as we vary the parameter $p$ which we use to vary the degree of distribution shift, because it is the exact form of the distribution $P_t(X, Y, V)$ that decides about the intrinsic difficulty of the regression or classification problem. On the other hand, the source predictors $P_s(Y|\cdot)$ are trained on the source joint distribution $P_s(X, Y, V)$ and evaluated on the samples from the target joint distribution $P_t(X, Y, V)$, and so they exhibit both degradation due to the distribution shift as well as the variation in performance due to the change in the intrinsic difficulty of the target distribution. Considering $P_t(Y|\cdot)$ and $P_s(Y|\cdot)$ in tandem allows us to disentangle both effects.

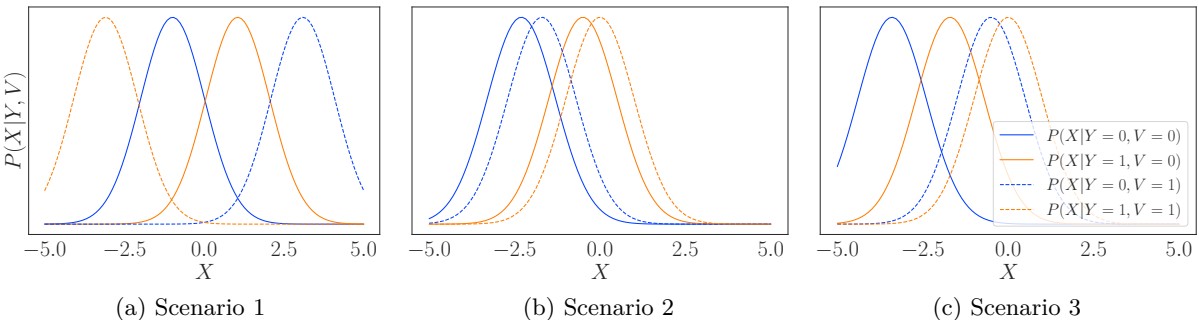

Figure 2: The four likelihood conditionals $P(X|Y,V)$, one for each combination of $Y, V$, for all three scenarios.

We also compare the performance of those predictors to a representative of the invariant learning methods mentioned in Section 1.1—a predictor $P_m(Y|X)$ trained according to the method introduced by Makar et al. (2022). This method involves learning a representation of covariate $X$ such that it is distributed independently of the covariate which is assumed to be spuriously correlated with $Y$, in our case $V$, and then forming a predictor based on that learned representation. Since this method entails a learned, rather than analytically derived, predictor, we perform a sweep over the relevant training hyperparameters. In order to consider the most optimistic case for the $P_m(Y|X)$ baseline's performance, we consider the hyperparameter setting most optimal according to the area under the accuracy-vs-$P_t(V)$ curve, which would have typically not be available as a selection criterium during the training procedure. In each of the figures, we present the mean and standard deviation over 100 training runs with the best hyperparameter setting. For details, see Appendix B.

The predictors $P_s(Y|\cdot)$ are computed analytically while the invariant predictor $P_m(Y|X)$ has to be learned empirically, which raises the question—is there a gap in performance between the analytical predictor and the empirically learned ones? In Appendix C, we show that for all three scenarios the empirically learned predictors $P_s^l(Y|\cdot)$ corresponding to the closed-form $P_s(Y|\cdot)$, with hyperparameters chosen in the same fashion as for $P_m(Y|X)$, reach performance very close to that of $P_s(Y|\cdot)$.

Even for a model as simple as Eq. (3), we can obtain qualitatively different behaviors for different choices of the parameters of the data generating process. Below, we consider two illustrative scenarios for the model in Eq. (3) and then consider the behavior of the presented estimators in the setting of a spuriously correlated features model as per $\mathcal{P}_{spur}$ in Eq. (1) and Fig. 1a.

### 3.1 Scenario 1: $P(Y|X,V)$ recovers lacking robustness of $P(Y|X)$ for shifts in $\mathcal{P}_{cause}$

Firstly, we consider a case where the performance of the predictor $P_s(Y|X)$ is severely impacted by the distribution shift, and, as the result, the robust predictor $P_s(Y|X,V)$ can shine. We choose the parameter values resulting in likelihoods presented in Fig. 2a: $P(Y=0|V=0)=0.2$, $P(Y=1|V=1)=0.9$, $\mu_{0,0}=-1, \mu_{1,0}=1, \mu_{0,1}=3, \mu_{1,1}=-3$.

In this setting, there are two effects worth paying attention to. Firstly, the relationship between $X$ and $Y$ is opposite for the two different values of $V$ in that for $V=0$, the larger the value of $X$, the higher the likelihood of $Y=1$, while for $V=1$, the lower the value of $X$, the higher the likelihood of $Y=1$. The predictor $P(Y|X)$, which does not have access to the value of $V$, is not able to account for this effect. Secondly, for $V=1$ (dashed), the conditionals for $Y=0$ and $Y=1$ are separated more than when $V=0$ (solid), hence making the classification problem $P(Y|X,V)$ easier.

In Fig. 3, we compare the performance of the predictors $P_s(Y|X)$ and $P_s(Y|X,V)$ across the family of distributions $\mathcal{P}_{cause}$, where the marginal $P_t(V=1)$ has been shifted w.r.t. the source distribution marginal $P_s(V=1)=0.4$. As predicted by theory, $P_s(Y|X,V)$'s performance does not degrade w.r.t. the optimal $P_t(Y|X,V)$ in neither accuracy, nor AUC, as $P_t(V)$ shifts further away from $P_s(V)$. On the other hand, the performance of $P_s(Y|X)$ degrades w.r.t. the optimal $P_t(Y|X)$ in terms of both accuracy and AUC, even performing worse than a random predictor for strong distribution shift at $p > 0.8$. Again, note that the

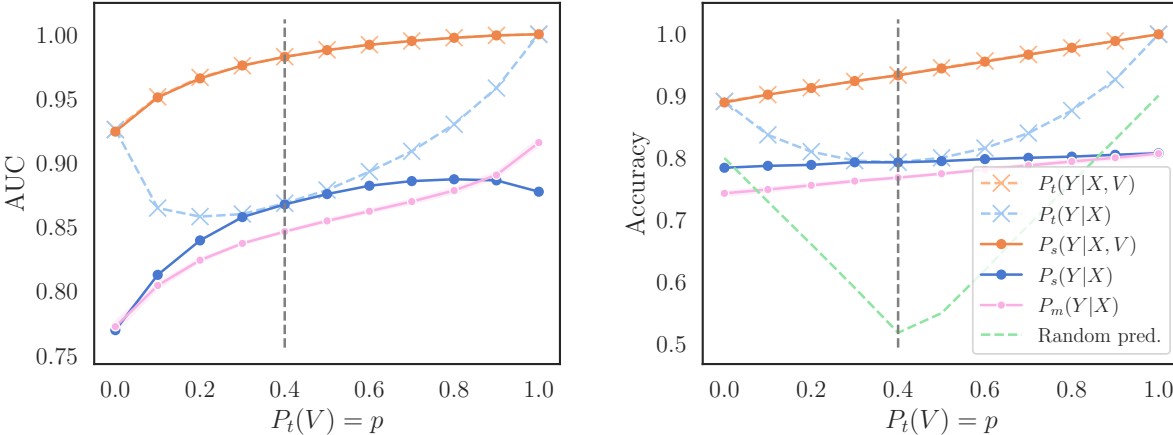

Figure 3: Scenario 1: AUC (left) and accuracy (right) of the models $P_s(Y|X)$ and $P_s(Y|X,V)$, the oracle predictors $P_t(Y|X)$ and $P_t(Y|X,V)$, and the invariant predictor $P_m(Y|X)$ of Makar et al. (2022), as a function of the target marginal $P_t(V) = p$, with source $P_s(V) = 0.4$ (grey dashed vertical line). Note that $P_s(Y|X,V)$ and $P_t(Y|X,V)$ almost perfectly overlap, up to the stochasticity of the estimation procedure. For $P_m(Y|X)$, we present the mean and std. dev. of the best hyperparameter setting according to the area under the accuracy-vs-$P_t(V)$ curve over 100 training runs. See the main text for details.

focus here is not on the comparison of the absolute performance between the predictors $P.(Y|X,V)$ and $P.(Y|X)$, but on the difference between the empirical performance of the trained predictors $P_s(Y|\cdot)$ and their corresponding optimal predictors $P_t(Y|\cdot)$—this gap quantifies the loss of performance of a given predictor due to distribution shift. The absolute performance of individual predictors $P(Y|\cdot)$ is changing due to the change in intrinsic difficulty of the classification problem, as discussed at the beginning of Section 3. In comparison, the invariant predictor $P_m(Y|X)$ of Makar et al. (2022) may perform better or worse than $P_s(Y|X)$ for varying distribution shift, but generally lacks robustness w.r.t. $P_t(Y|X)$, similarly to $P_s(Y|X)$.

### 3.2 Scenario 2: Marginal difference in robustness of $P(Y|X,V)$ vs. $P(Y|X)$ for shifts in $\mathcal{P}_{cause}$

Next, we consider a scenario where the effect of the distribution shift on the performance of the standard predictor $P(Y|X)$ is limited, even if the shift follows $\mathcal{P}_{cause}$. This can happen when knowing $V$ in addition to $X$ does not help the predictive performance to begin with, e.g., because under all distribution shifts the effect of $Y$ on $X$ is much more pronounced than the effect of $V$ on $Y$, or in the trivial case when $V$ can be expressed as a deterministic function of $X$ and therefore $P(Y|X,V) = P(Y|X)$. In our simulation, the former case can be achieved with parameter values: $P(Y=0|V=0)=0.15$, $P(Y=1|V=1)=0.3$, $\mu_{0,0}=-1, \mu_{1,0}=0.92$, $\mu_{0,1}=0.07$, $\mu_{1,1}=1.38$, $P_s(V=1)=0.5$. We display the likelihoods $P(X|Y,V)$ for this scenario in Fig. 2b.

We present the results for this scenario in Fig. 4. In this scenario, while the generated shifts do belong to $\mathcal{P}_{cause}$ and one of the generative model's parameters varies significantly, there is little to no difference in downstream predictive performance, even under strong distribution shift. What is more, the performance of the optimal predictors $P_t(Y|X)$ and $P_t(Y|X,V)$ is almost identical, and so is the performance of all the other predictors, including the invariant predictor $P_m(Y|X)$, which perform at a level close to the theoretical optimum through the entire spectrum of the distribution shift. Still, in general, while not improving over $P_s(Y|X)$, using $P_s(Y|X,V)$ does not hurt the performance even in this scenario, as indicated by the theoretical results in Section 2, as long as the shifts under consideration remain in the family $\mathcal{P}_{cause}$.

### 3.3 Scenario 3: Neither $P(Y|X,V)$ nor $P(Y|X)$ robust for shifts in $\mathcal{P}_{spur}$, in general

In the two previous scenarios considered, we saw that for shifts of $\mathcal{P}_{cause}$, the use of predictor $P(Y|X,V)$ instead of $P(Y|X)$ can lead to large benefits in distribution shift robustness, and even if that is not the case,

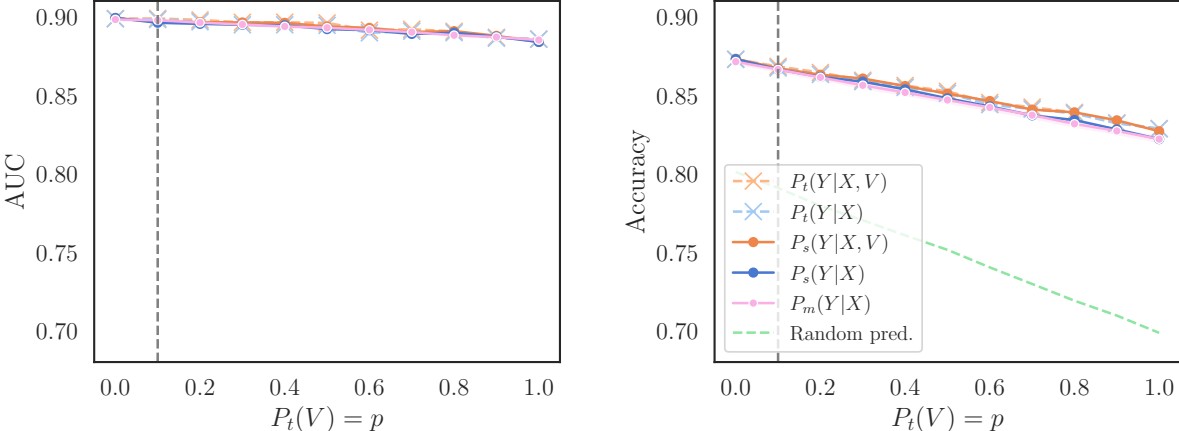

Figure 4: Scenario 2: AUC (left) and accuracy (right) of the models $P_s(Y|X)$ and $P_s(Y|X,V)$, the oracle predictors $P_t(Y|X)$ and $P_t(Y|X,V)$, and the invariant predictor $P_m(Y|X)$ of Makar et al. (2022), as a function of the target marginal $P_t(V) = p$, with source $P_s(V) = 0.1$ (grey dashed vertical line). For $P_m(Y|X)$, we present the mean and std. dev. of the best hyperparameter setting according to the area under the accuracy-vs-$P_t(V)$ curve over 100 training runs. See the main text for details.

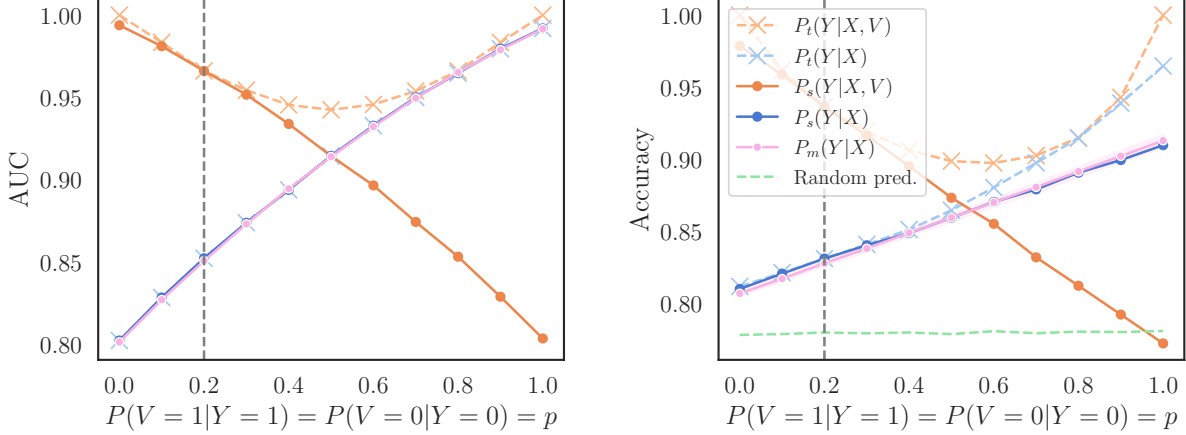

Figure 5: Scenario 3: AUC (left) and accuracy (right) of the models $P_s(Y|X)$ and $P_s(Y|X,V)$, the oracle predictors $P_t(Y|X)$ and $P_t(Y|X,V)$, and the invariant predictor $P_m(Y|X)$ of Makar et al. (2022), as a function of the distribution shift under $\mathcal{P}_{spur}$: $P_t(V = 1|Y = 1) = P_t(V = 0|Y = 0) = p$, with source $P_s(V = 1|Y = 1) = P_s(V = 0|Y = 0) = 0.2$ (grey dashed vertical line). For $P_m(Y|X)$, we present the mean and std. dev. of the best hyperparameter setting according to the area under the accuracy-vs-$P_t(V)$ curve over 100 training runs. See the main text for details.

it does not hurt the performance. However, what happens if the underlying shift does not belong to $\mathcal{P}_{cause}$, but rather to $\mathcal{P}_{spur}$? To investigate that question, we adjust the generative model in Eq. (3) such that it follows the distribution shift of the family $\mathcal{P}_{spur}$, where $P(V|Y)$ can be shifted by varying $p$ while $P(Y)$ and $P(X|Y,V)$ will remain the same:

$$P(Y = 1) = p_2, \qquad P(V = 0|Y = 0) = p,$$
$$P(X|Y = y, V = v) = \mathcal{N}(\mu_{y,v}, 1), \qquad P(V = 1|Y = 1) = p. \tag{4}$$

Note that we keep $P(V = 0|Y = 0) = P(V = 1|Y = 1) = p$ such that we can easily vary the degree of the distribution shift using a single parameter $p$, drawing inspiration from Makar et al. (2022).

We present the results for one representative instantiation of this model in Fig. 5. The parameters used are $P(Y = 1) = 0.22$, $\mu_{0,0} = -3.4$, $\mu_{1,0} = -0.5$, $\mu_{0,1} = -1.7$, $\mu_{1,1,} = 0.0$, $P_s(V = 0|Y = 0) = P_s(V = 1|Y = 1) = 0.2$, and we display the likelihoods corresponding to this model in Fig. 2c. The most important observation is that neither $P_s(Y|X)$ nor $P_s(Y|X, V)$ are robust in this setting. Depending on the exact parameters of the model, the source distribution parameter $p$, and the degree of the distribution shift, it is possible for either of the two predictors to perform better than the other. In this setting, both predictors might exhibit a significant gap between the source and the target variant, unlike in the $\mathcal{P}_{cause}$ family. Also, in this setting, there is no theoretical guarantee on the superiority of $P(Y|X, V)$ and so if the shifts belong to $\mathcal{P}_{spur}$, conditioning the predictor on $V$ can lead to results that are less robust than for $P(Y|X)$. Lastly, as expected, in this particular setting, the invariant predictor $P_m(Y|X)$ is slightly more robust than $P_s(Y|X)$.

### 3.4 Discussion

The predictive performance of a predictor that learns $P(Y|X)$ on the source distribution can be strongly impaired by a distribution shift in $P(V)$ in the anti-causal setting. For the data generating process from Fig. 1b, as showcased in Section 3.1, using a predictor that learns $P(Y|X, V)$ can recover the desired robustness.

On the other hand, there exist scenarios in which the distribution shift follows the setting from Fig. 1b, but the resulting performance improvement is negligible (Section 3.2). Still, in this setting, using $P(Y|X, V)$ does not hurt the performance in comparison to $P(Y|X)$.

Lastly, as expected, using a predictor $P(Y|X, V)$ can harm performance if the data generating process follows the spurious setting $\mathcal{P}_{spur}$ (Eq. (1) & Fig. 1a). Therefore, it is important to characterize the data generating process and the shift as well as possible to choose an appropriate predictor.

## 4 Experiments on health data

In this section, we conduct experiments on real, publicly available health data, the PTB-XL data set of annotated ECG recordings (Wagner et al., 2020). We induce synthetic distribution shifts of varying strength in the test sets. Our goal is to showcase how the proposed theory and the learnings from the simulation study can be applied in practice. Firstly, we verify that the shift we induce belongs to $\mathcal{P}_{cause}$, rather than the $\mathcal{P}_{spur}$ family. Then, supported by the theory, we add $V$ to the model input. As shown by the simulation study, we know that the impact on performance may vary depending on whether this combination of data generating process and induced distribution shift resembles more Scenario 1 or Scenario 2. In this case, the results yield evidence that we are in Scenario 2 and adding $V$ improves the performance only marginally and not consistently, however as expected without hurting the performance compared to $P(Y|X)$.

### 4.1 Experimental setup

**Data**  We use the PTB-XL data set (Wagner et al., 2020). It contains 21,837 clinical 12-lead ECG recordings from 18,885 patients. Each recording is 10 seconds long and is processed following previous literature at a frequency of 100 Hz (Strodthoff et al., 2021). Each ECG recording is annotated with an ECG statement that can be grouped into 5 classes of superdiagnostics: normal ECG (NORM), conduction disturbance (CD), myocardial infarction (MI), hypertrophy (HYP), and ST/T changes (STTC). We set up our experiment as a multi-label binary classification problem for the 4 superdiagnostic categories that indicate abnormal ECG. Furthermore, the data set contains an annotation of patient's age for 21,748 of the recordings, rounded to integer values. We use these annotations as a $V$ variable.

**Model and training**  We use the best performing model in the supervised deep learning benchmark analysis by Strodthoff et al. (2021), a xresnet1d101 model. We use the publicly available `tsai` library (Oguiza, 2022) to implement the xresnet1d101 model and obtain test set performance that is similar to the one reported in Strodthoff et al. (2021) (using the same training, validation, and test splits). When incorporating the integer

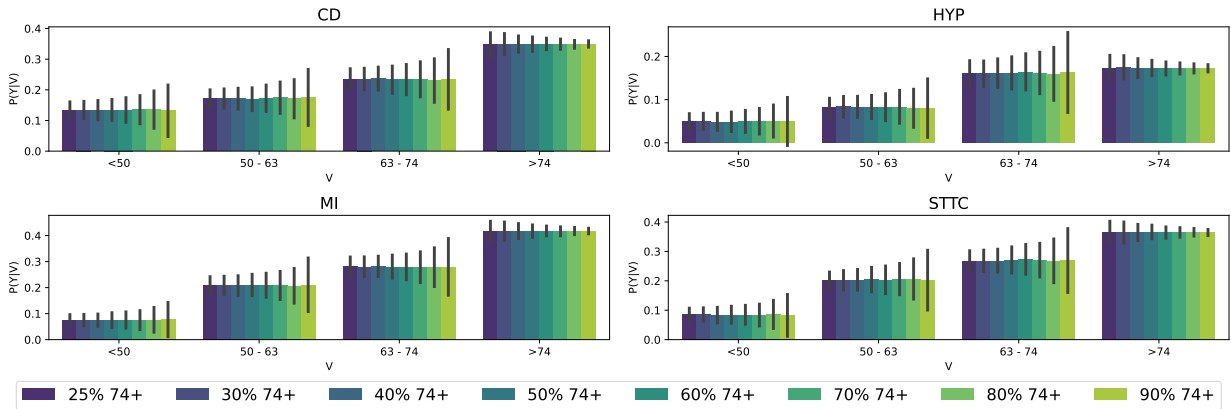

Figure 6: Validation of shifts for V=Age. For each of the 4 superdiagnostics (subplots), we show $P(Y|V_A = v)$ per $v \in \{< 50, \ 50 - 63, \ 63 - 74, \ 74+\}$ (leftmost group of bars in subplot to rightmost group of bars in subplot) for the original, unshifted test set (dark purple) and the shifts introduced in Section 4.1. Equal height of bars inside each group of bars is indicative for constant $P(Y|V)$ across evaluation sets.

valued $V$ as input to the predictors $P(Y|X, V)$, we use FiLM (Perez et al., 2018) to account for the influence of $V$. We explored other methods of incorporating information about $V$ (e.g., concatenating its value to the last layer representations before the linear mapping to logits) but found them to be less effective than FiLM.

**Inducing distribution shifts**    We induce a distribution shift in the test set by changing $P(V)$ in comparison to the original test set, which follows the same distribution as the training set. To this end, we discretise age such that $V_A \in \{< 50, \ 50–63, \ 63–74, \ 74+\}$ serves as a demographic variable. These age thresholds were chosen to yield sufficient samples per bin, and approximately correspond to the quartiles of the age distribution. We deliberately sample subsets of the test set such that the subsets contain a specific percentage of data points belonging to the highest age quartile, i.e., above 74 years. We increase this percentage from 25% (corresponding to the original training and test distributions) to 90%. For example, the shifted evaluation set '70% 74+' consists of 70% of samples from the test set from people that are over 74, and 30% from people that are at most 74 years old. We account for the resulting variance in the results by running evaluations over $10,000$ independently sampled subsets, and reporting averages and standard deviations.

### 4.2   Verifying that shifts belong to $\mathcal{P}_{cause}$ rather than $\mathcal{P}_{spur}$

As discussed in Section 3.4, it is important to first verify that the (target) distributions resulting from the shifts induced belong to the family $\mathcal{P}_{cause}$, i.e., that while $P(V)$ changes, $P(Y|V)$ and $P(X|Y, V)$ remain unchanged. In our case, we have access to some amount of $V, Y$ data from the target domain at the time of the development of the system. In this case, we can verify that $P(Y|V)$ remains unchanged by counting. In Fig. 6, we verify that indeed $P_t(Y|V_A) \approx P_s(Y|V_A)$ for all $t$ under inspection. Here, the source distribution $s$ is the one of the subsampled according to the original training and test distribution, i.e., with 25% of the samples being older than 74 years (column '25% 74+'). We can see that for all the target distributions $t$ introduced by the shifts in Section 4.1 (remaining columns), $P(Y|V_A = v)$ remains roughly constant for all $v \in \{< 50, \ 50–63, \ 63–74, \ 74+\}$. Also, the difference between $P(Y|V_A = v_i)$ and $P(Y|V_A = v_j)$ remains prominent for all $v_i \neq v_j \in \{< 50, \ 50–63, \ 63–74, \ 74+\}$ for all superdiagnostics and all shifts. As a result, we can exclude Scenario 3 (Section 3.3).

Of course, data from the target domain is not always available. However in the health domain, in many cases medical domain knowledge can be used to determine that $P(Y|V)$ remains unchanged, even when data from the target domain is not available. For example, we might know that the probability of a heart condition ($Y$) given patient age ($V$) is the same for country A (source) and country B (target).

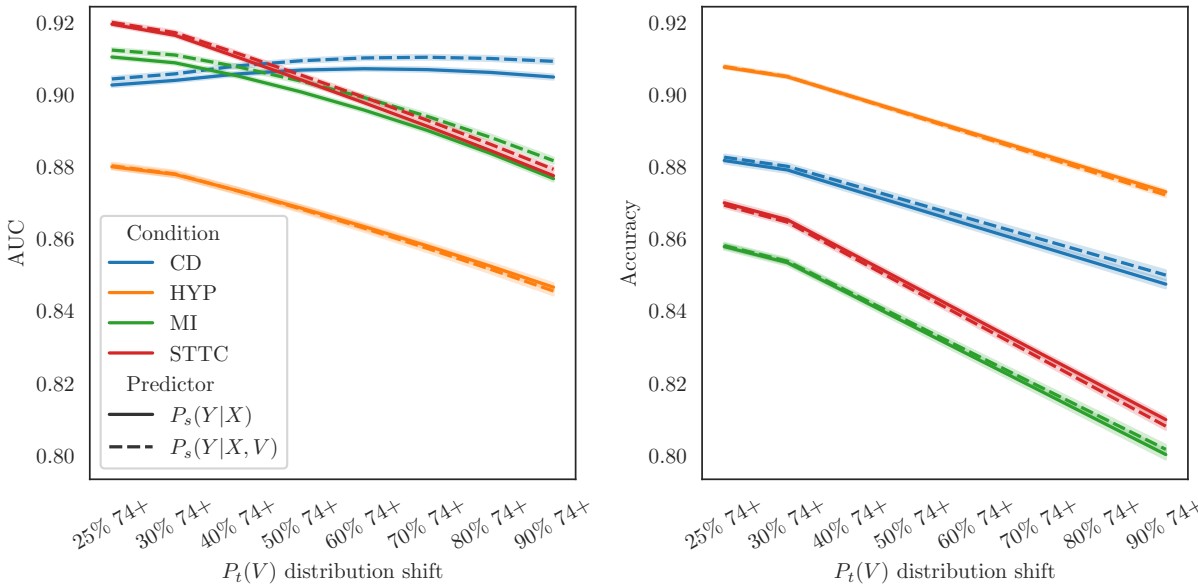

Figure 7: Comparison of the performance of predictors $P_s(Y|X)$ and $P_s(Y|X,V)$ on the PTB-XL dataset in terms of AUC (left) and accuracy (right), as a function of the target marginal $P_t(V)$, with source $P_s(V)$ at 25% 74+, corresponding to the entire training dataset. Random predictor accuracy baseline omitted for clarity: for HYP, starting at 0.88 decreasing to 0.84, and for all others, starting at $< 0.80$ and decreasing to $< 0.70$.

## 4.3 Results

As shown in the simulation study in Section 3, the impact of including $V$ into the model may vary even when $\mathcal{P}_{cause}$ applies. In Scenario 1, adding $V$ will make the predictor indeed more robust, while in Scenario 2, adding $V$ will not improve the robustness but it will not hurt it either.

Fig. 7 shows the area under the ROC curve (AUC) and predictive accuracy under the shifts in the family $\mathcal{P}_{cause}$, comparing the performance of the learned predictors $P_s(Y|X)$ and $P_s(Y|X,V)$. Unlike in the simulation setting, with this dataset we cannot compare to oracle predictors $P_t(Y|\cdot)$, as for the shifted distributions the available subsets of the training set become too small (around 3500 points) to reasonably train our xresnet101 on $P_t$ directly. Thus, both methods here are trained on the full training set and evaluated by the mean of the performance metrics they achieve on 10,000 draws of the shifted evaluation sets for each training run. Mean and standard errors over 100 training runs are shown.

With the exception of AUC for the CD superdiagnostic, for both predictors $P_s(Y|X)$ and $P_s(Y|X,Y)$, and for all remaining superdiagnostic categories, we observe a decrease in AUC and accuracy relative to a model evaluated on the source population ('25% 74+').

We observe that to some extent, and in particular for MI and CD, performance under shift can be slightly improved by incorporating $V$ into our prediction, but overall, the relative gains are not very pronounced, and not consistent throughout all superdiagnostic categories. Importantly, however, as predicted including $V$ hurts neither original nor shifted test set AUC. This complies with our theory and the results of the simulation study, and yields evidence that for this shift on PTB-XL, we find ourselves in Scenario 2. We tested whether this is due to the fact that $V$ can be perfectly predicted from $X$, but found that this is not the case—setting up a regression problem of predicting age $V$ using signal $X$ allowed to infer $V$ only up to 64% of explained variance. Similar as in Section 3.2, the reason may be that the effect of $Y$ on $X$ is much stronger than that of $V$ on $Y$, and at the same time that relationship is not changed much by changes in $V$. Finally, since the theoretical considerations in Section 2 only concern population quantities, our estimates based on finite samples may not inherit their robustness properties due to estimation errors. For example, it is possible that our model has not learned the conditioning on $V$ correctly.

## 5  Conclusion

In anti-causal prediction problems for diagnostic health applications, beyond the observation $X$ (e.g., ECG, X-ray), we often have additional covariates $V$ (e.g., patient metadata) at our disposal. In many existing works applying machine learning to health, these are ignored, or even used to make predictors invariant to them. However, we show that in anti-causal settings in which covariates $V$ causally influence the outcome of interest $Y$, rather than being spuriously related to them, $P(Y|X)$ in general does not remain stable across shifts in $P(V)$, while $P(Y|X, V)$ does. As such, regressing $Y$ only on $X$ to learn $P(Y|X)$, or invariant variations thereof, will lead to predictions that might not be robust under such shifts, while regressing $Y$ on $X$ and $V$ leads to the desired robustness. In a simulation study we identify both a scenario in which this is particularly important, as well as a scenario in which the effect on robustness is negligible. Importantly, we demonstrate that as long as the distribution shift does not belong to the spurious setting as per Fig. 1a, learning $P(Y|X, V)$ can improve, but not decrease robustness over learning $P(Y|X)$. Lastly, we analyze a real world healthcare application of predicting diagnostic statements from ECG recordings. We show that the (synthetically) induced shifts of patient age distributions indeed fall into the family $\mathcal{P}_{cause}$. Thus, our theory and synthetic simulation predict that we can safely include the auxiliary covariates. We confirm this also for this particular dataset where the gain is limited but the robustness is not weakened (akin to Scenario 2). We hope for our work to raise awareness when auxiliary covariates are useful to make machine learning deployment in healthcare more robust to distribution shifts.

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

# A  Proof of Proposition 1

First, we show that for any element $P_t$ of this family, it holds that $P_t(Y|X,V) = P_s(Y|X,V)$. Remember that by the definition of $\mathcal{P}_{cause}$, we have that $P_t(X \mid Y, V) = P_s(X \mid Y, V)$ and $P_t(Y \mid V) = P_s(Y \mid V)$. From this, it quickly follows that

$$P_t(X|V) = \int P_t(X|Y,V)P_t(Y|V)dY \tag{5}$$

$$= \int P_s(X|Y,V)P_s(Y|V)dY \tag{6}$$

$$= P_s(X|V) \tag{7}$$

Then, using basic probability calculus, it follows

$$P_t(Y|X,V) = \frac{P_t(Y,X,V)}{P_t(X,V)} \tag{8}$$

$$= \frac{P_s(X|Y,V)P_t(Y|V)P_t(V)}{P_t(X|V)P_t(V)} \tag{9}$$

$$= \frac{P_s(X|Y,V)P_s(Y|V)}{P_t(X|V)} \tag{10}$$

$$= \frac{P_s(X|Y,V)P_s(Y|V)}{P_s(X|V)} \tag{11}$$

$$= \frac{P_s(X|Y,V)P_s(Y|V)P_s(V)}{P_s(X|V)P_s(V)} \tag{12}$$

$$= P_s(Y|X,V) \tag{13}$$

Next, we show that for such an element $P_t$ of this family, in general $P_t(Y|X) \neq P_s(Y|X)$. Using the above result, this is indeed easy to see when marginalising over $V$:

$$P_t(Y|X) = \int P_t(Y|X,V)P_t(V|X)dV \tag{14}$$

$$= \int P_s(Y|X,V)P_t(V|X)dV \tag{15}$$

$$= \int P_s(Y|X,V)\frac{P_t(X|V)P_t(V)}{P_t(X)}dV \tag{16}$$

$$= \int P_s(Y|X,V)P_s(X|V)\frac{P_t(V)}{P_t(X)}dV \tag{17}$$

$$\tag{18}$$

Since in general $\frac{P_t(V)}{P_t(X)} \neq \frac{P_s(V)}{P_s(X)}$, this also implies that in general $P_t(Y|X) \neq P_s(Y|X)$.

Similarly,

$$P_t(Y) = \int\int P_s(X|Y,V)P_s(Y|V)P_t(V)dXdV \tag{19}$$

and since in general $P_t(V) \neq P_s(V)$, this also implies that in general $P_t(Y) \neq P_s(Y)$.

## B   Details on learning predictors empirically

We train the predictors with stochastic optimization using AdamW optimizer (Loshchilov & Hutter, 2019) with batch size $2^{14}$ for 1024 gradient steps, with most training runs plateauing in terms of loss value at latest around 150 gradient steps. Each sample and each batch are independently sampled from the assumed data generating process.

We perform a random search sweep over the learning rate (range: $10^{-2.5}$–$10^1$) and the MMD regularization coefficient $\alpha$ (range: $10^{-10}$–$10^0$) using 2048 randomly sampled values, and present the results for the hyperparameter setting with the highest area under the accuracy-vs-$P_t(V)$ curve.

## C   Comparison of the predictors: closed-form $P_s(Y|\cdot)$ vs. empirically learned $P_s^l(Y|\cdot)$

In all three cases, Figs. 8 to 10, we see that the performance of $P_s^l(Y|\cdot)$ follows the performance of closed-form predictor $P_s(Y|\cdot)$, and we conclude that indeed the empirically learned predictors can match the performance of the closed-form predictors. In the case of Scenario 3 (Fig. 10), we notice a slight divergence between the learned and the analytical predictor of $P(Y|X,V)$ as the distribution shift becomes more extreme, which is likely due to slight underfitting of $P_s^l(Y|X,V)$ on the source distribution that it benefits from as the spuriousity is reversed more extremely. However, the performance of $P_s^l(Y|X,V)$ qualitatively still follows the one of $P_s(Y|X,V)$ even for those more extreme shifts, confirming the conclusions drawn in Section 3.3.

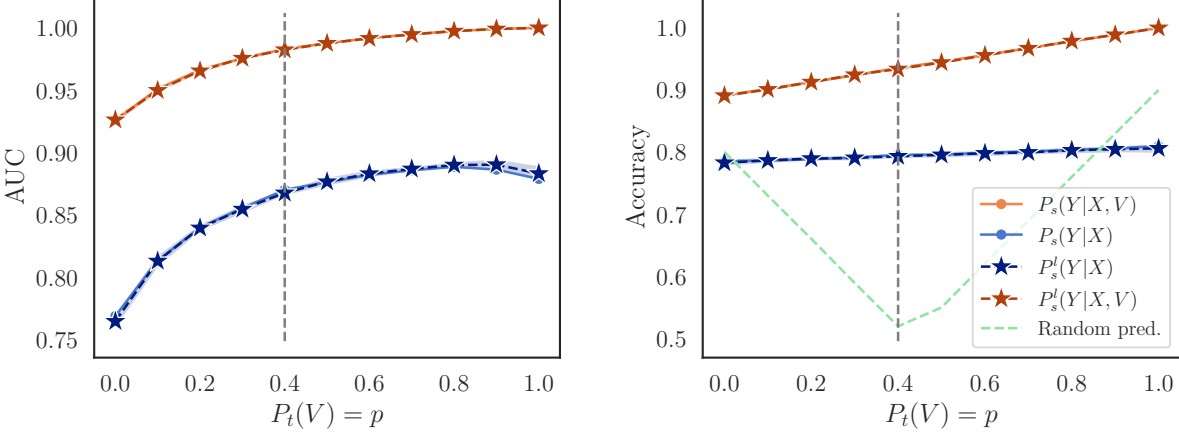

Figure 8: Scenario 1: AUC (left) and accuracy (right) of the closed-form predictors $P_s(Y|X)$ and $P_s(Y|X,V)$, and the empirically learned predictors $P_s^l(Y|X)$ and $P_s^l(Y|X,V)$, as a function of the target marginal $P_t(V) = p$, with source $P_s(V) = 0.4$ (grey dashed vertical line). For $P_s^l(Y|\cdot)$, we display the standard deviation over 10 lowest-training-loss runs varying random seed and hyperparameters.

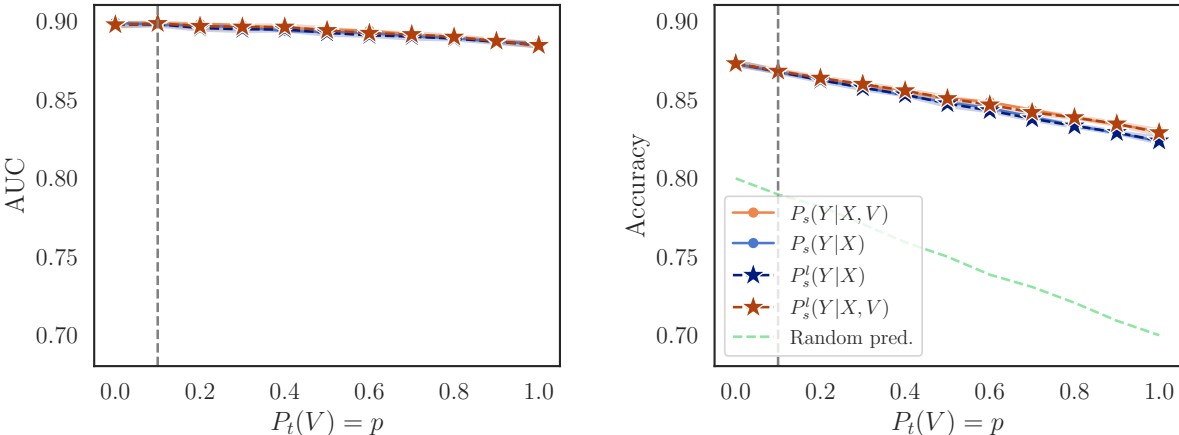

Figure 9: Scenario 2: AUC (left) and accuracy (right) of the closed-form predictors $P_s(Y|X)$ and $P_s(Y|X,V)$, and the empirically learned predictors $P_s^l(Y|X)$ and $P_s^l(Y|X,V)$, as a function of the target marginal $P_t(V) = p$, with source $P_s(V) = 0.1$ (grey dashed vertical line). For $P_s^l(Y|\cdot)$, we display the standard deviation over 10 lowest-training-loss runs varying random seed and hyperparameters.

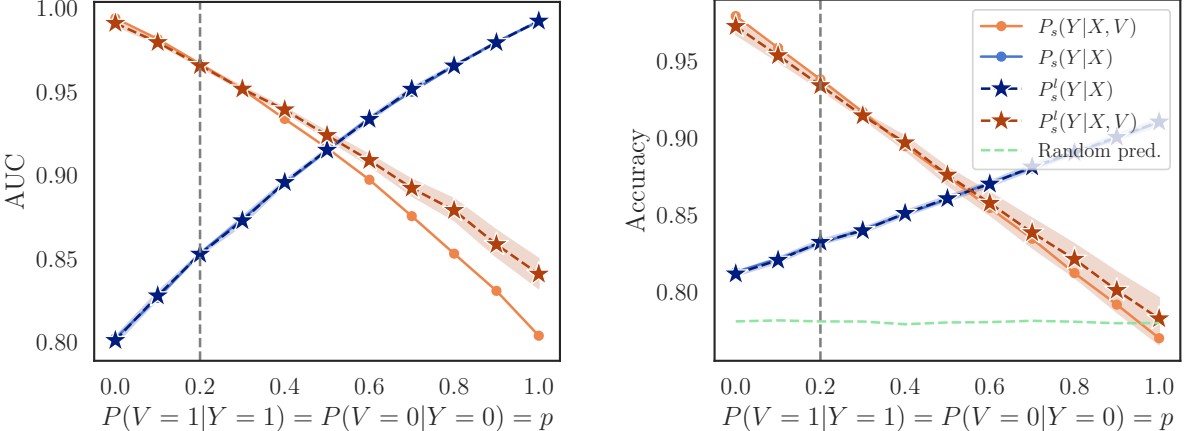

Figure 10: Scenario 3: AUC (left) and accuracy (right) of the closed-form predictors $P_s(Y|X)$ and $P_s(Y|X,V)$, and the empirically learned predictors $P_s^l(Y|X)$ and $P_s^l(Y|X,V)$, as a function of the distribution shift under $\mathcal{P}_{spur}$: $P_t(V = 1|Y = 1) = P_t(V = 0|Y = 0) = p$, with source $P_s(V = 1|Y = 1) = P_s(V = 0|Y = 0) = 0.2$ (grey dashed vertical line). For $P_s^l(Y|\cdot)$, we display the standard deviation over 10 lowest-training-loss runs varying random seed and hyperparameters.

