# OpenReview forum: "Considerations for Distribution Shift Robustness of Diagnostic Models in Healthcare"
_TMLR — Rejected by TMLR_

### Review · Reviewer_PeoN · 2024-07-02

**Summary Of Contributions:**

The authors present a set of synthetic and semi-synthetic experiments illustrating that model mismatch can lead to poor performance in distribution shift problems.

**Audience:**

Yes

**Broader Impact Concerns:**

None.

**Claims And Evidence:**

No

**Requested Changes:**

Following from the above weaknesses, I recommend:
1. Using non-synthetic distribution shifts to illustrate the different settings described in the paper and to provide evidence that these models are relevant to the medical ML community.
2. Update the medical examples used and provide evidence by way of citations that they are a match for the corresponding models.
3. Expand 4.2 to handle realistic settings.

**Strengths And Weaknesses:**

## Strengths:
1. I found the paper clear and well-written.
2. The simulations in Section 3 are clear and illustrative.
3. Experiments are well-described and appear to be reproducible.
4. As best I can tell, all claims appear technically correct.

## Weaknesses:
1. I think this paper makes for a strong tutorial on model mismatch, but I'm not sure what the scientific contribution is here. Proposition 1 follows from very basic independence properties of graphical models (to be fair, the authors do not claim otherwise). Section 3 is an illustrative, but simplistic demonstration of these basic independence properties. Finally, it is not clear to me what is meant to be learned from Section 4 since the distribution shift is synthetic.
2. It is also not clear to me how these demonstrations are specific to medicine. The authors assert, without real evidence, that these types of shifts are common in medicine. Certainly the fully synthetic experiments have nothing really to do with clinical risk prediction. In reality, the causal structures encountered in clinical data are extremely complex and high dimensional and involve many layers of time-varying confounding, missingness, and measurement error. One potential way to strengthen the paper would be to examine some datasets with real distribution shifts and make the case for one or the other model.
3. As best I can tell, Section 4.2 is only possible because of the low dimensional setting the authors are working with. In reality, most biomarkers will depend a range of patient variables and this type of visual would not be feasible. Another potentially interesting direction would be to provide methods for testing this in realistic high-dimensional settings.
4. In Section 1.1, the authors should avoid saying that BMI causes LVH. BMI is caused by (among other things) adiposity, which itself is related to, but not always a direct cause of many conditions like LVH. In the case of BMI and LVH, specifically, it is my understanding that the mechanism is unclear and may actually have to do with blood pressure rather than body fat directly. In reality, the relationship between BMI and LVH may be more like Figure 1a. I suggest that the authors select their examples with care so as to avoid the same model mismatch that the paper illustrates.

---

### Review · Reviewer_UDHn · 2024-07-04

**Summary Of Contributions:**

This paper studies methods to make diagnostic models robust to changes in patient demographics, which is crucial when applying models across different settings. It highlights the necessity of incorporating auxiliary covariates like age or BMI to enhance model robustness against these distribution shifts. Through theoretical insights, extensive simulations, and real-world data application using the PTB-XL dataset, the study demonstrates that models including demographic information withstand these shifts better. The authors suggest that machine learning practitioners in healthcare carefully model data processes to decide if including auxiliary covariates is beneficial in their specific settings. This work could influence future research and practices in machine learning applications for healthcare.

**Audience:**

Yes

**Broader Impact Concerns:**

No ethical implications involved.

**Claims And Evidence:**

Yes

**Requested Changes:**

It is recommended that authors update their papers based on the aforementioned weaknesses.

**Strengths And Weaknesses:**

Strengths:

1. This paper contains extensive experimental results, spanning from synthetic data to data collected in real-world applications, which could be crucial for the future development of this domain.

2. Authors highlighted the importance of metadata $V$ in mitigating and measuring the distribution shift.

3. Authors give a rather comprehensive discussion on different scenarios of the effect of distribution shift.

Weaknesses:

1. Some crucial empirical details seem to be missing; since the authors incorporated both metadata and biomarkers in the modeling process, how are these different types of data modeled altogether? It appears to be a mixture of dense data (biomarker) and sparse data (metadata); what techniques are used to make sure that they are exploited effectively?

2. proposition 1 seems to be a direct result of the assumption (Equ. 2); therefore, the non-triviality and importance of Proposition 1 need to be highlighted under the context.

3. It seems that the distribution shift is only applied to a single variable, which is the age of the population; this is potentially insufficient for proving the authors' points.

4. Overall, the biggest concern I hold is that the argument made in this paper is somewhat obvious and does not bear much insight - **authors are simply claiming that if there exists a distribution shift on metadata $V$, then introducing $V$ during the predictive process is generally beneficial - which is sort of obvious and expected**.

5. There are no baselines compared in this paper; intuitively, some relevant works in handling distribution shift could be naively migrated to this problem; not comparing with those methods undermines the empirical significance of this paper.

Minor issues:

1. The layout of the proof does not accord with typical standards; the authors recommend updating the layout of the proof based on some example papers.

---

### Review · Reviewer_33jm · 2024-07-07

**Summary Of Contributions:**

This paper considers a classification setting under distribution shift in the context of healthcare, i.e., predict the outcome $Y$ from a biomarker $X$ with the existence of a covariate $V$ and the distribution shift is caused by the change of the covariate $V$. Different from the spurious relation modeling in previous studies, the authors propose a new modeling for data generation process, i.e., causal relation between $V$ and $Y$. Then the authors conduct a simulation study, and show that when data belongs to the causal relation modeling, ignoring covariates as well as common invariant learning approaches cannot not yield robust predictors in the studied setting. Instead, considering the covariate as additional input can obtain a relatively robust predictors. Experiments are also conducted in real applications to confirm their claim.

**Audience:**

Yes

**Broader Impact Concerns:**

Not applicable.

**Claims And Evidence:**

No

**Requested Changes:**

Please refer to the weakness 1, 2, and 6.

**Strengths And Weaknesses:**

Strength:

1. It is worthy of praise to remind ML practitioners in the healthcare domain, to analyze and select the appropriate ML models in their specific setting. (in this paper, carefully analyze whether including auxiliary covariates is appropriate in their setting).

2. It is nice to discuss under what scenarios using $P(Y|X,V)$ as the predictor will (or will not) benefit the prediction performance (Section 3).

3. The authors empirically verify whether the data shift belongs to the causal relation in Section 4.2, which would be a guidance for readers how to decide which relation the data shift belongs to.

Weakness:

1. This paper lacks powerful experiments in real applications to support the effectiveness of the predictor $P(Y|X,V)$, which significantly hurt its application value. The experimental results for real-data show that $P(Y|X,V)$ improves prediction performance only marginally and not consistently. It is highly desirable to find several real-world datasets to show that $P(Y|X,V)$ leads to an obvious increase in performance.

2. The authors are expected to show the universality of the proposed causal relationship scenario (Figure 1(b)) in healthcare domain. The authors should demonstrate at least three real-world datasets belongs to such scenario. Otherwise, it’s hard to make the ML practitioners in healthcare to attach importance to the insight in this paper.

3. I have several questions about the probabilistic modeling of the Spurious relation in Eq.(1).

(1) It hard for me to understand $P(V|Y)$. Since there is no causal relationship between $V$ and $Y$, so why $Y$ would affect $X$?

(2) If such an effect $P(V|Y)$ is caused by the fictitious variable $C$ connecting $V$ and $Y$, then I think it’s a correlation instead of a causal relationship. This means that we can also consider $P(Y,V) = P(Y|V)P(V)$, which is the same with the causal relation in Eq. (2).

4. How can we distinguish between the three scenarios, from the parameter values? For example, in section 3.1, why the set of parameter value indicates that the predictor $P_s(Y |X)$ is severely impacted by the distribution shift?

5. According to Figure 5, there is little difference between $P_s(Y|X)$ and $P_m(Y|X)$. It seems that the invariant learning does not help relieve the distribution shift problem. Could you please give more explanations?

6. Only a set of parameter values is used for simulation experiment, which is not enough. It is expected that more simulation experiments are con

---

### Decision · Action_Editor_KYsz · 2024-09-16

**Recommendation:** Reject

**Comment:**

This paper constructs causal graphs to model the underlying generative processes of healthcare data for medical diagnosis. The main contribution lies in the introduction of the variable \(V\), which captures patient background information. The authors then propose methods to improve prediction reliability when \(V\) is subject to external interventions, leading to distributional changes.

While the theory and methodology are sound and correct under the given assumptions, the reviewers raised two major concerns. First, the theoretical framework is a direct application of existing results, such as independence properties in graphical models. Second, although the paper focuses on modeling healthcare data, there is no evidence that the model’s assumptions hold in real-world data. The experiments use synthetic distribution shifts based on model assumptions rather than actual distribution shifts in real data.

While the first concern does not violate TMLR’s acceptance criteria, the second concern was not adequately addressed in the rebuttal. Therefore, I recommend rejecting the paper for now. However, I encourage the authors to resubmit to TMLR after addressing the practical aspects of their method, ideally demonstrating its validity in at least one real healthcare application.

**Audience:**

Yes

**Claims And Evidence:**

no.

There is no strong evidence that the assumptions made in this paper match the real distributions shift scenarios in healthcare.

**Resubmission Of Major Revision:**

The authors may consider submitting a major revision at a later time.